# Principal Neighbourhood Aggregation for Graph Nets

**Gabriele Corso**[*]
University of Cambridge
gc579@cam.ac.uk

**Luca Cavalleri**[*]
University of Cambridge
lc737@cam.ac.uk

**Dominique Beaini**
InVivo AI
dominique@invivoai.com

**Pietro Liò**
University of Cambridge
pietro.lio@cst.cam.ac.uk

**Petar Veličković**
DeepMind
petarv@google.com

## Abstract

Graph Neural Networks (GNNs) have been shown to be effective models for different predictive tasks on graph-structured data. Recent work on their expressive power has focused on isomorphism tasks and countable feature spaces. We extend this theoretical framework to include continuous features—which occur regularly in real-world input domains and within the hidden layers of GNNs—and we demonstrate the requirement for multiple aggregation functions in this context. Accordingly, we propose Principal Neighbourhood Aggregation (PNA), a novel architecture combining multiple aggregators with degree-scalers (which generalize the sum aggregator). Finally, we compare the capacity of different models to capture and exploit the graph structure via a novel benchmark containing multiple tasks taken from classical graph theory, alongside existing benchmarks from real-world domains, all of which demonstrate the strength of our model. With this work, we hope to steer some of the GNN research towards new aggregation methods which we believe are essential in the search for powerful and robust models.

## 1 Introduction

Graph Neural Networks (GNNs) have been an active research field for the last ten years with significant advancements in graph representation learning [1, 2, 3, 4]. However, it is difficult to understand the effectiveness of new GNNs due to the lack of standardized benchmarks [5] and of theoretical frameworks for their expressive power.

In fact, most work in this domain has focused on improving the GNN architectures on a set of graph benchmarks, without evaluating the capacity of their network to accurately characterize the graphs' structural properties. Only recently there have been significant studies on the expressive power of various GNN models [6, 7, 8, 9, 10]. However, these have mainly focused on the isomorphism task in domains with countable features spaces, and little work has been done on understanding their capacity to capture and exploit the underlying properties of the graph structure.

We hypothesize that the aggregation layers of current GNNs are unable to extract enough information from the nodes' neighbourhoods in a single layer, which limits their expressive power and learning abilities.

We first mathematically prove the need for multiple aggregators and propose a solution for the uncountable multiset injectivity problem introduced by [6]. Then, we propose the concept of degree-scalers as a generalization to the *sum* aggregation, which allow the network to amplify or attenuate signals based on the degree of each node. Combining the above, we design the proposed *Principal*

---

[*]Equal contribution.

*Neighbourhood Aggregation* (PNA) model and demonstrate empirically that multiple aggregation strategies improve the performance of the GNN.

Dehmamy *et al.* [11] have also empirically found that using multiple aggregators (mean, sum and normalized mean), which extract similar statistics from the input message, improves the performance of GNNs on the task of graph moments. In contrast, our work extends the theoretical framework by deriving the necessity to use complementary aggregators. Accordingly, we propose the use of different statistical aggregations to allow each node to better understand the distribution of the messages it receives, and we generalize the *mean* as the first of a set of possible *n-moment* aggregators. In the setting of graph kernels, Cai *et al.* [12] constructed a simple baseline using multiple aggregators. In the field of computer vision, Lee *et al.* [13] empirically showed the benefits of combining *mean* and *max* pooling. These give us further confidence in the validity of our theoretical analysis.

We present a consistently well-performing and parameter efficient encode-process-decode architecture [14] for GNNs. This differs from traditional GNNs by allowing a variable number of convolutions with shared parameters. Using this model, we compare the performances of some of the most diffused models in the literature (GCN [15], GAT [16], GIN [6] and MPNN [17]) with our PNA.

Previous work on tasks taken from classical graph theory focuses on evaluating the performance of GNN models on a single task such as shortest paths [18, 19, 20], graph moments [11] or travelling salesman problem [5, 21]. Instead, we took a different approach by developing a multi-task benchmark containing problems both on the node level and the graph level. Many of the tasks are based on dynamic programming algorithms and are, therefore, expected to be well suited for GNNs [19]. We believe this multi-task approach ensures that the GNNs are able to understand multiple properties simultaneously, which is fundamental for solving complex graph problems. Moreover, efficiently sharing parameters between the tasks suggests a deeper understanding of the structural features of the graphs. Furthermore, we explore the generalization ability of the networks by testing on graphs of larger sizes than those present in the training set.

To further demonstrate the performance of our model, we also run tests on recently proposed real-world GNN benchmark datasets [5, 22] with tasks taken from molecular chemistry and computer vision. Results show the PNA outperforms the other models in the literature in most of the tasks hence further supporting our theoretical findings.

The code for all the aggregators, scalers, models (in PyTorch, DGL and PyTorch Geometric frameworks), architectures, multi-task dataset generation and real-world benchmarks is available here.

## 2 Principal Neighbourhood Aggregation

In this section, we first explain the motivation behind using multiple aggregators concurrently. We then present the idea of degree-based scalers, linking to prior related work on GNN expressiveness. Finally, we detail the design of graph convolutional layers which leverage the proposed Principal Neighbourhood Aggregation.

### 2.1 Proposed aggregators

Most work in the literature uses only a single aggregation method, with *mean*, *sum* and *max* aggregators being the most used in the state-of-the-art models [6, 15, 17, 18]. In Figure 1, we observe how different aggregators fail to discriminate between different messages when using a single GNN layer.

We formalize our observations in the theorem below:

**Theorem 1** (Number of aggregators needed). *In order to discriminate between multisets of size $n$ whose underlying set is $\mathbb{R}$, at least $n$ aggregators are needed.*

**Proposition 1** (Moments of the multiset). *The moments of a multiset (as defined in Equation 4) exhibit a valid example using $n$ aggregators.*

We prove Theorem 1 in Appendix A and Proposition 1 in Appendix B. Note that unlike Xu *et al.* [6], we consider a continuous input feature space; this better represents many real-world tasks where the observed values have uncertainty, and better models the latent node features within a neural network's representations. Continuous features make the space uncountable, and void the injectivity proof of the *sum* aggregation presented by Xu *et al.* [6].

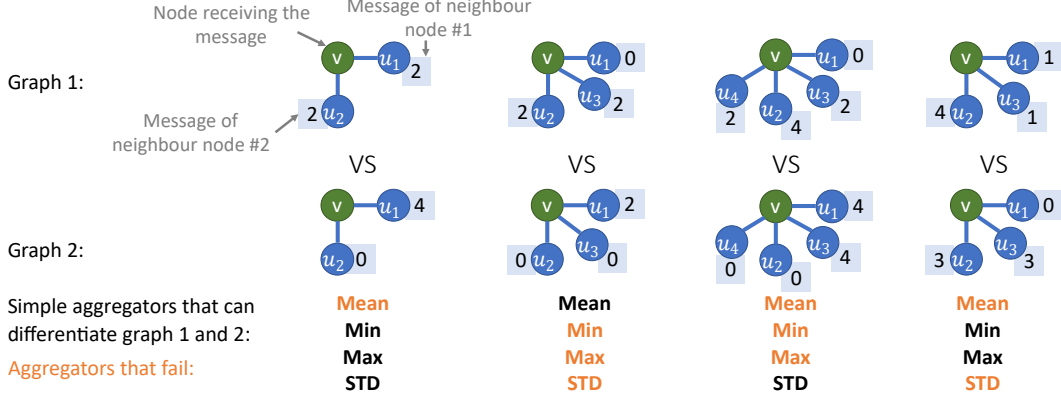

Figure 1: Examples where, for a single GNN layer and continuous input feature spaces, some aggregators fail to differentiate between neighbourhood messages.

Hence, we redefine aggregators as continuous functions of multisets which compute a statistic on the neighbouring nodes, such as *mean*, *max* or *standard deviation*. The continuity is important with continuous input spaces, as small variations in the input should result in small variations of the aggregators' output.

Theorem 1 proves that the number of independent aggregators used is a limiting factor of the expressiveness of GNNs. To empirically demonstrate this, we leverage four aggregators, namely *mean*, *maximum*, *minimum* and *standard deviation*. Furthermore, we note that this can be extended to the *normalized moment* aggregators, which allow advanced distribution information to be extracted whenever the degree of the nodes is high.

The following paragraphs will describe the aggregators we leveraged in our architectures.

**Mean aggregation** $\mu(X^l)$   The most common message aggregator in the literature, wherein each node computes a weighted average or sum of its incoming messages. Equation 1 presents, on the left, the general mean equation, and, on the right, the direct neighbour formulation, where $X$ is any multiset, $X^l$ are the nodes' features at layer $l$, $N(i)$ is the neighbourhood of node $i$ and $d_i = |N(i)|$. For clarity we use $\mathbb{E}[f(X)]$ where $X$ is a multiset of size $d$ to be defined as $\mathbb{E}[f(X)] = \frac{1}{d} \sum_{x \in X} f(x)$.

$$\mu(X) = \mathbb{E}[X] \quad , \quad \mu_i(X^l) = \frac{1}{d_i} \sum_{j \in N(i)} X_j^l \tag{1}$$

**Maximum and minimum aggregations** $\max(X^l),\ \min(X^l)$   Also often used in literature, they are very useful for discrete tasks, for domains where credit assignment is important and when extrapolating to unseen distributions of graphs [18]. Alternatively, we present the softmax and softmin aggregators in Appendix E, which are differentiable and work for weighted graphs, but don't perform as well on our benchmarks.

$$\max_i(X^l) = \max_{j \in N(i)} X_j^l \quad , \quad \min_i(X^l) = \min_{j \in N(i)} X_j^l \tag{2}$$

**Standard deviation aggregation** $\sigma(X^l)$   The standard deviation (STD or $\sigma$) is used to quantify the spread of neighbouring nodes features, such that a node can assess the diversity of the signals it receives. Equation 3 presents, on the left, the standard deviation formulation and, on the right, the STD of a graph-neighbourhood. *ReLU* is the rectified linear unit used to avoid negative values caused by numerical errors and $\epsilon$ is a small positive number to ensure $\sigma$ is differentiable.

$$\sigma(X) = \sqrt{\mathbb{E}[X^2] - \mathbb{E}[X]^2} \quad , \quad \sigma_i(X^l) = \sqrt{ReLU\left(\mu_i(X^{l^2}) - \mu_i(X^l)^2\right) + \epsilon} \tag{3}$$

**Normalized moments aggregation** $M_n(X^l)$   The mean and standard deviation are the first and second normalized moments of the multiset ($n = 1, n = 2$). Additional moments, such as the

skewness ($n = 3$), the kurtosis ($n = 4$), or higher moments, could be useful to better describe the neighbourhood. These become even more important when the degree of a node is high because four aggregators are insufficient to describe the neighbourhood accurately. As described in Appendix D, we choose the n$^{th}$ root normalization, as presented in Equation 4, because it gives a statistic that scales linearly with the size of the individual elements (as the other aggregators); this gives the training adequate numerical stability. Once again we add an $\epsilon$ to the absolute value of the expectation before applying the n$^{th}$ root for numerical stability of the gradient.

$$M_n(X) = \sqrt[n]{\mathbb{E}\left[(X - \mu)^n\right]} \ , \ \ n > 1 \tag{4}$$

## 2.2 Degree-based scalers

We introduce scalers as functions of the number of messages being aggregated (usually the node degree), which are multiplied with the aggregated value to perform either an *amplification* or an *attenuation* of the incoming messages.

Xu *et al.* [6] show that the use of *mean* and *max* aggregators by themselves fail to distinguish between neighbourhoods with identical features but with differing cardinalities, and the same applies to all the aggregators described above. They propose the use of the *sum* aggregator to discriminate between such multisets. We generalise their approach by expressing the *sum* aggregator as the composition of a *mean* aggregator and a linear-degree amplifying scaler $S_{\mathrm{amp}}(d) = d$.

**Theorem 2** (Injective functions on countable multisets). *The mean aggregation composed with any scaling linear to an injective function on the neighbourhood size can generate injective functions on bounded multisets of countable elements.*

We formalize and prove Theorem 2 in Appendix C; the results proven in [6] about the *sum* aggregator become then a particular case of this theorem, and we can use any kind of injective scaler to discriminate between multisets of various sizes.

Recent work shows that summation aggregation doesn't generalize well to unseen graphs [18], especially when larger. One reason is that a small change of the degree will cause the message and gradients to be amplified/attenuated exponentially (a linear amplification at each layer will cause an exponential amplification after multiple layers). Although there are different strategies to deal with this problem, we propose using a logarithmic amplification $S \propto \log(d + 1)$ to reduce this effect. Note that the logarithm is injective for positive values, and $d$ is defined non-negative.

Further motivation for using logarithmic scalers is to better describe the neighbourhood influence of a given node. Suppose we have a social network where nodes A, B and C have respectively 5 million, 1 million and 100 followers: on a linear scale, nodes B and C are closer than A and B; however, this does not accurately model their relative influence. This scenario exhibits how a logarithmic scale can discriminate better between messages received by *influencer* and *follower* nodes.

We propose the logarithmic scaler $S_{\mathrm{amp}}$ presented in Equation 5, where $\delta$ is a normalization parameter computed over the training set, and $d$ is the degree of the node receiving the message.

$$S_{\mathrm{amp}}(d) = \frac{\log(d + 1)}{\delta} \quad , \quad \delta = \frac{1}{|\mathrm{train}|} \sum_{i \in \mathrm{train}} \log(d_i + 1) \tag{5}$$

We further generalize this scaler in Equation 6, where $\alpha$ is a variable parameter that is negative for attenuation, positive for amplification or zero for no scaling. Other definitions of $S(d)$ can be used—such as a linear scaling—as long as the function is injective for $d > 0$.

$$S(d, \alpha) = \left(\frac{\log(d + 1)}{\delta}\right)^{\alpha}, \quad d > 0, \quad -1 \le \alpha \le 1 \tag{6}$$

## 2.3 Combined aggregation

We combine the aggregators and scalers presented in previous sections obtaining the Principal Neighbourhood Aggregation (PNA). This is a general and flexible architecture, which in our tests we used with four neighbour-aggregations with three degree-scalers each, as summarized in Equation 7.

The aggregators are defined in Equations 1–3, while the scalers are defined in Equation 6, with $\otimes$ being the tensor product.

$$\bigoplus = \underbrace{\begin{bmatrix} I \\ S(D, \alpha = 1) \\ S(D, \alpha = -1) \end{bmatrix}}_{\text{scalers}} \otimes \underbrace{\begin{bmatrix} \mu \\ \sigma \\ \max \\ \min \end{bmatrix}}_{\text{aggregators}} \tag{7}$$

As mentioned earlier, higher degree graphs such as social networks could benefit from further aggregators (e.g. using the moments proposed in Equation 4). We insert the PNA operator within the framework of a message passing neural network [17], obtaining the following GNN layer:

$$X_i^{(t+1)} = U\left(X_i^{(t)}, \bigoplus_{(j,i)\in E} M\left(X_i^{(t)}, E_{j\to i}, X_j^{(t)}\right)\right) \tag{8}$$

where $E_{j\to i}$ is the feature (if present) of the edge $(j, i)$, $M$ and $U$ are neural networks (for our benchmarks, a linear layer was enough). $U$ reduces the size of the concatenated message (in space $\mathbb{R}^{13F}$) back to $\mathbb{R}^F$ where $F$ is the dimension of the hidden features in the network. As in the MPNN paper [17], we employ multiple towers to improve computational complexity and generalization performance.

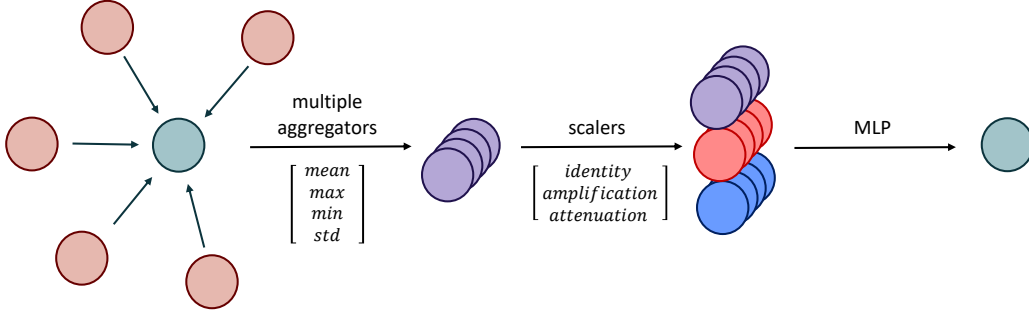

Figure 2: Diagram for the Principal Neighbourhood Aggregation or PNA.

Using twelve operations per kernel will require the usage of additional weights per input feature in the $U$ function, which could seem to be just quantitatively—not qualitatively—more powerful than an ordinary MPNN with a single aggregator [17]. However, the overall increase in parameters in the GNN model is modest and, as per our theoretical analysis above, a limiting factor of GNNs is likely their usage of a single aggregation.

This is comparable to convolutional neural networks (CNN) where a simple $3 \times 3$ convolutional kernel requires 9 weights per feature (1 weight per neighbour). Using a CNN with a single weight per $3 \times 3$ kernel will reduce the computational capacity since the feedforward network won't be able to compute derivatives or the Laplacian operator. Hence, it is intuitive that the GNNs should also require multiple weights per node, as previously demonstrated in Theorem 1. In Appendix K, we will demonstrate this observation empirically, by running experiments on baseline models with larger dimensions of the hidden features (and, therefore, more parameters).

## 3 Architecture

We compare the performance of the PNA layer against some of the most popular models in the literature, namely GCN [15], GAT [16], GIN [6] and MPNN [17] on a common architecture. In Appendix F, we present the details of these graph convolutional layers.

For the multi-task experiments, we used an architecture, represented in Figure 3, with $\mathcal{M}$ convolutions followed by three fully-connected layers for node labels and a set2set (S2S)[23] readout function for graph labels. In particular, we want to highlight:

**Gated Recurrent Units (GRU)** [24] applied after the update function of each layer, as in [17, 25]. Their ability to retain information from previous layers proved effective when increasing the number of convolutional layers $\mathcal{M}$.

**Weight sharing** in all the GNN layers but the first makes the architecture follow an encode-process-decode configuration [3, 14]. This is a strong prior which works well on all our experimental tasks, yields a parameter-efficient architecture, and allows the model to have a variable number $\mathcal{M}$ of layers.

**Variable depth** $\mathcal{M}$, decided at inference time (based on the size of the input graph and/or other heuristics), is important when using models over high variance graph distributions. In our experiments we have only used heuristics dependant on the number of nodes $N$ ($\mathcal{M} = f(N)$) and, for the architectures in the results below, we settled with $\mathcal{M} = \lfloor N/2 \rfloor$. It would be interesting to test heuristics based on properties of the graph, such as the diameter, or an adaptive computation time heuristic [26, 27] based on, for example, the convergence of the nodes features [18]. We leave these analyses to future work.

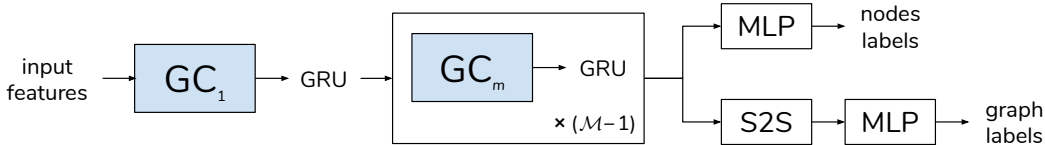

Figure 3: Layout of the architecture used. When comparing different models, the difference lies only in the type of graph convolution used in place of $GC_1$ and $GC_m$.

This architecture layout was chosen for its performance and parameter efficiency. We note that all architectural attempts yielded similar comparative performance of GNN layers and in Appendix I we provide the results for a more *standard* architecture.

## 4   Multi-task benchmark

The benchmark consists of classical graph theory tasks on artificially generated graphs.

**Random graph generation**    Following previous work [18, 28], the benchmark contains undirected unweighted randomly generated graphs of a wide variety of types. In Appendix G, we detail these types, and we describe the random toggling used to increase graph diversity. For the presented multi-task results, we used graphs of small sizes (15 to 50 nodes) as they were already sufficient to demonstrate clear differences between the models.

**Multi-task graph properties**    In the multi-task benchmark, we consider three node labels and three graph labels based on standard graph theory problems. The node properties tasks are the single-source shortest-path lengths, the eccentricity and the Laplacian features ($LX$ where $L = (D - A)$ is the Laplacian matrix and $X$ the node feature vector). The graph properties tasks are whether the graph is connected, the diameter and the spectral radius.

**Input features**    As input features, the network is provided with two vectors of size $N$, a one-hot vector (representing the source for the shortest-path task) and a feature vector $X$ where each element is i.i.d. sampled as $X_i \sim \mathcal{U}[0, 1]$. Apart from taking part in the Laplacian features task, this random feature vector also provides a *unique identifier* for the nodes in other tasks. Similar strengthening via random features was also concurrently discovered by [29]. This allows for addressing some of the problems highlighted in [7, 30]; e.g. the task of whether a graph is connected could be performed by continually aggregating the maximum feature of the neighbourhood and then checking whether they are all equal in the readout.

**Model training**    While having clear differences, these tasks also share related subroutines (such as graph traversals). While we do not take this sharing of subroutines as prior as in [18], we expect models to pick up on these commonalities and efficiently share parameters between the tasks, which reinforce each other during the training.

We trained the models using the Adam optimizer for a maximum of 10,000 epochs, using early stopping with a patience of 1,000 epochs. Learning rates, weight decay, dropout and other hyper-parameters were tuned on the validation set. For each model, we run 10 training runs with different seeds and different hyper-parameters (but close to the tuned values) and report the five with least validation error.

## 5    Results and discussion

### 5.1    Multi-task artificial benchmark

The multi-task results are presented in Figure 4a, where we observe that the proposed PNA model consistently outperforms state-of-the-art models, and in Figure 4b, where we note that the PNA performs better on all tasks. The *baseline* represents the MSE from predicting the average of the training set for all tasks.

The trend of these multi-task results follows and amplifies the difference in the average performances of the models when trained separately on the individual tasks. This suggests that the PNA model can better capture and exploit the common sub-units of these tasks. Appendix J provides the average results of the models when trained on individual tasks. Moreover, PNA showed to perform the best on all architecture layouts that we attempted (see Appendix I) and on all the various types of graphs (see Appendix H).

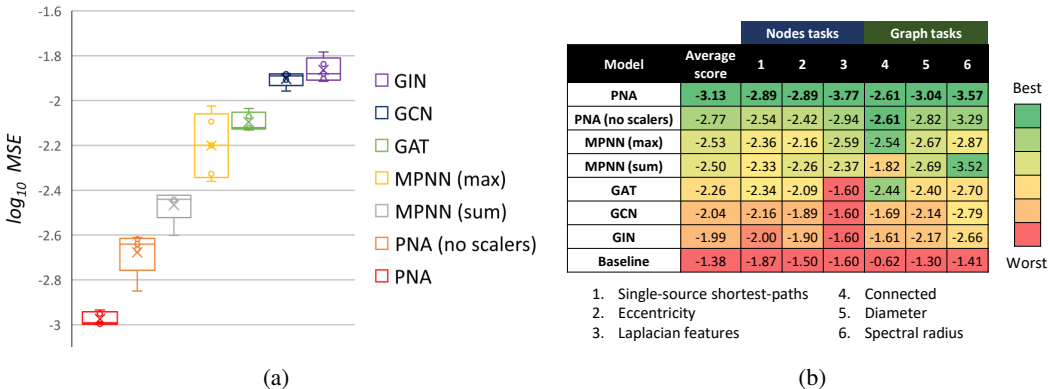

Figure 4: Multi-task benchmarks for different GNN models using the same architecture and various near-optimal hyper-parameters. (a) Distribution of the $\log_{10}$MSE errors for the top 5 performances of each model. (b) Mean $\log_{10}$MSE error for each task and their combined average.

To demonstrate that the performance improvements of the PNA model are not due to the (relatively small) number of additional parameters it has compared to the other models (about 15%), we ran tests on all the other models with latent size increased from 16 to 20 features. The results, presented in Appendix K, suggest that even when these models are given 30% more parameters than the PNA, they are qualitatively less capable of capturing the graph structure.

Finally, we explored the extrapolation of the models to larger graphs, in particular, we trained models on graphs of sizes between 15 and 25, validated between 25 and 30 and evaluate between 20 and 50. This task presents many challenges, two of the most significant are: firstly, unlike in [18] the models are not given any step-wise supervision or trained on easily extendable subroutines; secondly, the models have to cope with their architectures being augmented with further hidden layers than trained on, which can sometimes cause problems with rapidly increasing feature scales.

Due to the aforementioned challenges, as expected, the performance of the models (as a proportion of the baseline performance) gradually worsens, with some of them having feature explosions. However, the PNA model keeps consistently outperforming all the other models on all graph sizes. Our results also follow the findings in [18], i.e. that between single aggregators the *max* tends to perform best when extrapolating to larger graphs.

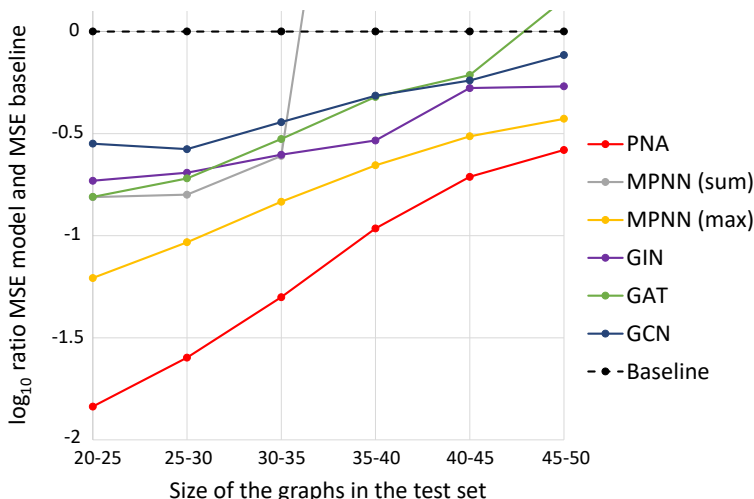

Figure 5: Multi-task $\log_{10}$ of the ratio of the MSE for different GNN models and the MSE of the baseline.

## 5.2 Real-world benchmarks

The recent works by Dwivedi *et al.* [5] and Hu *et al.* [22] have shown problems with many benchmarks used for GNNs in recent years and proposed a new range of datasets across different artificial and real-world tasks. To test the capacity of the PNA model in real-world domains, we assessed it on their chemical (ZINC and MolHIV) and computer vision (CIFAR10 and MNIST) datasets.

To ensure a fair comparison of the different convolutional layers, we followed their method for training procedure (data splits, optimizer, etc.) and GNN structure (layers, normalization and approximate number of parameters). For the MolHIV dataset, we used the same GNN structure as in [31].

| | | ZINC | | CIFAR10 | | MNIST | | MolHIV |
|---|---|---|---|---|---|---|---|---|
| | **Model** | No edge features | Edge features | No edge features | Edge features | No edge features | Edge features | No edge features |
| | | MAE | MAE | Acc | Acc | Acc | Acc | % ROC-AUC |
| | MLP | 0.710±0.001 | | 56.01±0.90 | | 94.46±0.28 | | |
| **Dwivedi** | GCN | 0.469±0.002 | | 54.46±0.10 | | 89.99±0.15 | | 76.06±0.97 |
| ***et al.*** | GIN | 0.408±0.008 | | 53.28±3.70 | | 93.96±1.30 | | 75.58±1.40 |
| **and Xu** | DiffPool | 0.466±0.006 | | 57.99±0.45 | | 95.02±0.42 | | |
| ***et al.*** | GAT | 0.463±0.002 | | 65.48±0.33 | | 95.62±0.13 | | |
| **papers** | MoNet | 0.407±0.007 | | 53.42±0.43 | | 90.36±0.47 | | |
| | GatedGCN | 0.422±0.006 | 0.363±0.009 | 69.19±0.28 | 69.37±0.48 | 97.37±0.06 | 97.47±0.13 | |
| | MPNN (sum) | 0.381±0.005 | 0.288±0.002* | 65.39±0.47 | 65.61±0.30 | 96.72±0.17 | 96.90±0.15 | |
| **Our** | MPNN (max) | 0.468±0.002 | 0.328±0.008* | 69.70±0.55 | **70.86±0.27** | 97.37±0.11 | 97.82±0.08 | |
| **experi-** | PNA (no scalers) | 0.413±0.006 | 0.247±0.036* | **70.46±0.44** | 70.47±0.72 | **97.41±0.16** | **97.94±0.12** | 78.76±1.04 |
| **ments** | PNA | **0.320±0.032** | **0.188±0.004*** | 70.21±0.15 | 70.35±0.63 | 97.19±0.08 | 97.69±0.22 | **79.05±1.32** |

Figure 6: Results of the PNA and MPNN models in comparison with those analysed by Dwivedi *et al.* and Xu *et al.* (GCN[15], GIN[6], DiffPool[32], GAT[16], MoNet[33] and GatedGCN[34]). * indicates the training was conducted with additional patience to ensure convergence.

To better understand the results in the table, we need to take into account how graphs differ among the four datasets. In the chemical benchmarks, graphs are diverse and individual edges (bonds) can significantly impact the properties of the graphs (molecules). This contrasts with computer vision datasets made of graphs with a regular topology (every node has 8 edges) and where the graph structure of the representation is not crucial (the good performance of the MLP is evidence).

With this and our theoretical analysis in mind, it is understandable why the PNA has a strong performance in the chemical datasets, as it was designed to understand the graph structure and better retain neighbourhood information. At the same time, the version without scalers suffers from the fact it cannot distinguish between neighbourhoods of different size. Instead, in the computer vision datasets the average improvement of the PNA on SOTA was lower due to the smaller importance of the graph structure and the version of the PNA without scalers performs better as the constant degree of these graphs makes scalers redundant (and it is better to 'spend' parameters for larger hidden sizes).

## 6  Conclusion

We have extended the theoretical framework in which GNNs are analyzed to continuous features and proven the need for multiple aggregators in such circumstances. We also have generalized the *sum* aggregation by presenting degree-scalers and proposed the use of a logarithmic scaling. Taking the above into consideration, we have presented a method, Principal Neighbourhood Aggregation, consisting of the composition of multiple aggregators and degree-scalers. With the goal of understanding the ability of GNNs to capture graph structures, we have proposed a novel multi-task benchmark and an encode-process-decode architecture for approaching it. Empirical results from synthetic and real-world domains support our theoretical evidence. We believe that our findings constitute a step towards establishing a hierarchy of models w.r.t. their expressive power, where the PNA model appears to outperform the prior art in GNN layer design.

## Broader Impact

Our work focuses mainly on theoretically analyzing the expressive power of Graph Neural Networks and can, therefore, play an indirect role in the (positive or negative) impacts that the field of graph representation learning might have on the domains where it will be applied.

More directly, our contribution in proving the limitations of existing GNNs on continuous feature spaces should help to provide an insight into their behaviour. We believe this is a significant result which might motivate future research aimed at overcoming such limitations, yielding more reliable models. However, we also recognize that, in the short-term, proofs of such weaknesses might spark mistrust against applications of these systems or steer adversarial attacks towards existing GNN architectures.

In an effort to overcome some of these short-term negative impacts and contribute to the search for more reliable models, we propose the Principal Neighbourhood Aggregation, a method that overcomes some of these theoretical limitations. Our tests demonstrate the higher capacity of the PNA compared to the prior art on both synthetic and real-world tasks; however, we recognize that our tests are not exhaustive and that our proofs do not allow for generating "optimal" aggregators for any task. As such, we do not rule out sub-optimal performance when applying the exact architecture proposed here to novel domains.

We propose the usage of aggregation functions, such as standard deviation and higher-order moments, and logarithmic scalers. To the best of our knowledge, these have not been used before in GNN literature. To further test their behaviour, we conducted out-of-distribution experiments, testing our models on graphs much larger than those in the training set. While the PNA model consistently outperformed other models and baselines, there was still a noticeable drop in performance. We therefore strongly encourage future work on analyzing the stability and efficacy of these novel aggregation methods on new domains and, in general, on finding GNN architectures that better generalize to graphs from unseen distributions, as this will be essential for the transition to industrial applications.

## Acknowledgements

The authors thank Saro Passaro for the valuable insights and discussion for the mathematical proofs.

## Funding Disclosure

Dominique Beaini is currently a Machine Learning Researcher at InVivo AI. Pietro Liò is a Full Professor at the Department of Computer Science and Technology of the University of Cambridge. Petar Veličković is a Research Scientist at DeepMind.

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
