[Supplementary Material]

## A Proof for Theorem 1 (Number of aggregators needed)

*In order to discriminate between multisets of size $n$ whose underlying set is $\mathbb{R}$, at least $n$ aggregators are needed.*

*Proof.* Let $S$ be the $n$-dimensional subspace of $\mathbb{R}^n$ formed by all tuples $(x_1, x_2, \ldots, x_n)$ such that $x_1 \leq x_2 \leq \ldots \leq x_n$, and notice how $S$ is the collection of the aforementioned multisets. We defined an aggregator as a continuous function from multisets to reals, which corresponds to a continuous function $g : S \to \mathbb{R}$.

Assume by contradiction that it is possible to discriminate between all the multisets of size $n$ using only $n-1$ aggregators, viz. $g_1, g_2, \ldots, g_{n-1}$.

Define $f : S \to \mathbb{R}^{n-1}$ to be the function mapping each multiset $X$ to its output vector $(g_1(X), g_2(X), \ldots, g_{n-1}(X))$. Since $g_1, g_2, \ldots, g_{n-1}$ are continuous, so is $f$, and, since we assumed these aggregators are able to discriminate between all the multisets, $f$ is injective.

As $S$ is a $n$-dimensional Euclidean subspace, it is possible to define a $(n-1)$-sphere $C^{n-1}$ entirely contained within it, i.e. $C^{n-1} \subseteq S$. According to Borsuk–Ulam theorem [35, 36], there are two distinct (in particular, non-zero and antipodal) points $\vec{x}_1, \vec{x}_2 \in C^{n-1}$ satisfying $f(\vec{x}_1) = f(\vec{x}_2)$, showing $f$ not to be injective; hence the required contradiction. $\square$

*Note: $n$ aggregators are actually sufficient.* A simple example is to use $g_1, g_2, \ldots, g_n$ where $g_k(X) = $ the $k$-th smallest item in $X$. It's clear to see that the multiset whose elements are $g_1(X), g_2(X), \ldots, g_n(X)$ is $X$, which can hence be uniquely determined by the aggregators.

## B Proof for Proposition 1 (Moments of the multiset)

*The moments of a multiset (as defined in Equation 4) exhibit a valid example using $n$ aggregators.*

*Proof.* Since $n \geq 1$, and the first aggregator is *mean*, we know $\mu$. Let $X = \{x_1, x_2, \ldots, x_n\}$ be the multiset to be found, and define $R = \{r_1 = x_1 - \mu, \ r_2 = x_2 - \mu, \ \ldots, \ r_n = x_n - \mu\}$.

Notice how $\sum r_i^1 = 0$, and for $1 < k \leq n$ we have $\sum r_i^k = n\, M_k(X)^k$, i.e. all the symmetric power sums $p_k = \sum r_i^k$ $(k \leq n)$ are uniquely determined by the moments.

Additionally, $e_k$, the elementary symmetric sums of $R$, i.e. the sum of the products of all the sub-multisets of size $k$ $(1 \leq k \leq n)$, are determined as follow:

$e_1$, the sum of all elements, is equal to $p_1$; $e_2$, the sum of the products of all pairs in $R$, is $(e_1 p_1 - p_2)/2$; $e_3$, the sum of the products of all triplets, is $(e_2 p_1 - e_1 p_2 + p_3)/3$, and so on. Notice how $e_1, e_2, \ldots, e_n$ can be computed using the following recursive formula [37]:

$$\sum_{1 \leq i_1 < i_2 < \cdots < i_k \leq n} \left( \prod_{j=1}^{k} r_{i_j} \right) = e_k = \frac{1}{k} \sum_{j=1}^{k} (-1)^{j-1} e_{k-j} p_j \quad , \quad e_0 = 1$$

Consider polynomial $P(x) = \Pi(x - r_i)$, i.e. the unique polynomial of degree $n$ with leading coefficient 1 whose roots are $R$. This defines $A$, the coefficients of $P$, i.e. the real numbers $a_0, a_1, \ldots, a_{n-1}$ for which $P(x) = x^n + a_{n-1}x^{n-1} + \ldots + a_1 x + a_0$. Using Vieta's formulas [38]:

$$\sum_{1 \leq i_1 < i_2 < \cdots < i_k \leq n} \left( \prod_{j=1}^{k} r_{i_j} \right) = (-1)^k \frac{a_{n-k}}{a_n}$$

we obtain

$$e_k = (-1)^k \frac{a_{n-k}}{a_n}$$
$$= (-1)^k a_{n-k} \qquad \text{recall } a_n = 1$$
$$\therefore a_i = (-1)^{n+i} e_{n+i} \qquad \text{letting } k = n + i \text{ and rearranging}$$

Hence $A$ is uniquely determined, and so is $P$, being its coefficients a valid definition of it. By the fundamental theorem of algebra, $P$ has $n$ (possibly repeated) roots, which are the elements of $R$, hence uniquely determining the latter.

Finally, $X$ can be easily determined adding $\mu$ to each element of $R$. □

*Note: the proof above assumes the knowledge of $n$.* In the case that $n$ is variable (as in GNNs), and so we have multisets of up to $n$ elements, an extra aggregator will be needed. An example of such aggregator is the *mean* multiplied by any injective scaler which would allow the degree of the node to be inferred.

## C  Proof for Theorem 2 (Injective functions on countable multisets)

*The mean aggregation composed with any scaling linear to an injective function on the neighbourhood size can generate injective functions on bounded multisets of countable elements.*

*Proof.* Let $\chi$ be the countable input feature space from which the elements of the multisets are taken and $X$ an arbitrary multiset. Since $\chi$ is countable and the cardinality of multisets is bounded, let $Z : \chi \to \mathbb{N}^+$ be an injection from $\chi$ to natural numbers, and $N \in \mathbb{N}$ such that $|X| + 1 < N$ for all $X$.

Let's define an injective function $s$, and without loss of generality, assume $s(0), s(1), \ldots, s(N) > 0$ (otherwise for the rest of the proof consider $s$ as $s'(i) = s(i) - \min_{j \in [0,N]} s(j) + \epsilon$ which is positive for all $i \in [0, N]$). $s(|X|)$ can only take value in $\{s(0), s(1), \ldots, s(N)\}$, therefore let us define $\gamma = \min\left\{ \frac{s(i)}{s(j)} \mid i, j \in [0, N],\ s(i) \geq s(j) \right\}$. Since $s$ is injective, $s(i) \neq s(j)$ for $i \neq j$, which implies $\gamma > 1$.

Let $K > \frac{1}{\gamma - 1}$ be a positive real number and consider $f(x) = N^{-Z(x)} + K$.

$\forall x \in \chi, Z(x) \in [1, N] \Rightarrow N^{-Z(x)} \in [0, 1] \Rightarrow f(x) \in [K, K+1]$, so $\mathbb{E}_{x \in X}[f(x)] \in [K, K+1]$.

We proceed to show that the cardinality of $X$ can be uniquely determined, and $X$ itself can be determined as well, by showing that exist an injection $h$ over the multisets.

Let us $h$ as a function that scales the mean of $f$ by an injective function of the cardinality:

$$h(X) = s(|X|)\, \mathbb{E}_{x \in X}[f(x)]$$

We want show that the value of $|X|$ can be uniquely inferred from the value of $h(X)$. Assume by contradiction $\exists\, X', X''$ multisets of size at most $N$ such that $|X'| \neq |X''|$ but $h(X') = h(X'')$; since $s$ is injective $s(|X'|) \neq s(|X''|)$, without loss of generality let $s(|X'|) > s(|X''|)$, then:

$$s(|X''|)(K+1) \geq s(|X''|)\, \mathbb{E}_{x \in X''}[f(x)] = h(X'') = h(X') = s(|X'|)\, \mathbb{E}_{x \in X'}[f(x)] \geq s(|X'|)\, K$$

$$\implies\ K \leq \frac{1}{\frac{s(|X'|)}{s(|X''|)} - 1} \leq \frac{1}{\gamma - 1}$$

which is a contradiction. So it is impossible for the size of a multiset $X$ to be ambiguous from the value of $h(X)$.

Let us define $d$ as the function mapping $h(X)$ to $|X|$.

$$h'(X) = \sum_{x \in X} N^{-Z(x)} = \frac{h(X)|X|}{s(|X|)} - K|X| = \frac{h(X)d(h(X))}{s(d(h(X)))} - Kd(h(X))$$

Considering the $Z(j)$-th digit $i$ after the decimal point in the base $N$ representation of $h'(X)$, it can be inferred that $X$ contains $i$ elements $j$, and, so, all the elements in $X$ can be determined; hence $h$ is injective over the multisets in $X$. □

*Note:* this proof is a generalization of the one by *Xu et al.* [6] on the *sum* aggregator.

## D   Normalized moments aggregation

The main motivation for choosing the $n^{th}$ root normalization for the moments is numerical stability. In fact, one property of our version is that it scales linearly with $L$, for uniformly distributed random variables $U[0, L]$, as do other aggregators such as mean, max and min (std is a particular case). Other common formulations of the moments such as those in Equation 9 scale respectively as the $n^{th}$ power and constantly with $L$. This difference causes numerical instability when combined in the same layer.

$$M_n(X) = \mathbb{E}\left[(X - \mu)^n\right] \qquad M_n(X) = \frac{\mathbb{E}\left[(X - \mu)^n\right]}{\sigma^n} \qquad (9)$$

To demonstrate the usefulness of higher moments aggregation and further motivate the need for multiple aggregation functions, we ran an ablation study showing how different moments affect the performance of the model. We conduct this by testing five different models, each taking a different number of moments, on our multi-task benchmark.

Figure 7: Multi-task $\log_{10}$ MSE on different versions of the PNA model with increasing number of moments aggregators (specified in the legend), using *mean* as first moment. All the models use the identity, amplification and attenuation scalers. The model on the right is the complete PNA as described before (*mean*, *max*, *min* and *std* aggregators).

The results in Figure 7 demonstrate that with the increase of the number of aggregators the models reach a higher expressive power, but at a certain point (dependent on the graphs and tasks, in this case around 3) the increase in expressiveness given by higher moments reduces the performance since the model becomes harder to optimize and prone to overfitting. We expect that higher moments will be more beneficial on graphs with a higher average degree since they will better characterize the neighbourhood distributions.

Finally, we note how the addition of the *max* and *min* aggregators in the PNA (rightmost column) gives a better and more consistent performance in these tasks than higher moments. We believe this is task-dependent, and, for algorithmic tasks, discrete aggregators can be valuable. As a side note, we point out how the *max* and *min* aggregators of positive values can be considered as the $n^{th}$-root of the $n^{th}$ (non-centralized) moment as n tends to, respectively, $+\infty$ and $-\infty$.

# E    Alternative aggregators

Besides those described above, we have experimented with additional aggregators. We detail some examples below. Domain-specific metrics can also be an effective choice.

**Softmax and softmin aggregations**    As an alternative to *max* and *min*, *softmax* and *softmin* are differentiable and can be weighted in the case of edge features or attention networks. They also allow an asymmetric message passing in the direction of the strongest signal. Equation 10 presents their direct neighbour formulations, where $X^l$ are the nodes features at layer $l$ with respect to node $i$ and $N(i)$ is the neighbourhood of node $i$:

$$\text{softmax}_i(X^l) = \sum_{j \in N(i)} \frac{X_j^l \exp(X_j^l)}{\sum_{k \in N(i)} \exp(X_k^l)} \quad , \quad \text{softmin}_i(X^l) = -\text{softmax}_i(-X^l) \tag{10}$$

# F    Alternative graph convolutions

In this section, we present the details of the four graph convolutional layers from existing models that we used to compare the performance of the PNA in the multi-task benchmark.

**Graph Convolutional Networks (GCN)**    [15] use a normalized mean aggregator followed by a linear transformation and an activation function. We define it in Equation 11, where $\tilde{A} = A + I_N$ is the adjacency matrix with self-connections, $W$ is a trainable weight matrix and $b$ a learnable bias.

$$X^{(t+1)} = \sigma\left(\tilde{D}^{-\frac{1}{2}}\tilde{A}\tilde{D}^{-\frac{1}{2}}X^{(t)}W + b\right) \tag{11}$$

**Graph Attention Networks (GAT)**    [16] perform a linear transformation of the input features followed by an aggregation of the neighbourhood as a weighted sum of the transformed features, where the weights are set by an attention mechanism $a$. We define it in Equation 12, where $W$ is a trainable projection matrix. As in the original paper, we employ the use of multi-head attention.

$$X_i^{(t+1)} = \sigma\left(\sum_{(j,i) \in E} a\left(X_i^{(t)}, X_j^{(t)}\right) W X_j^{(t)}\right) \tag{12}$$

**Graph Isomorphism Networks (GIN)**    [6] perform a sum aggregation over the neighbourhood, followed by an update function $U$ consisting of a multi-layer perceptron. We define it in Equation 13, where $\epsilon$ is a learnable parameter. As in the original paper, we use a 2-layer MLP for $U$.

$$X_i^{(t+1)} = U\left(\left(1 + \epsilon\right)X_i^{(t)} + \sum_{j \in N(i)} X_j^{(t)}\right) \tag{13}$$

**Message Passing Neural Networks (MPNN)**    [17] perform a transformation before and after an arbitrary aggregator. We define it in Equation 14, where $M$ and $U$ are neural networks and $\bigoplus$ is a single aggregator. In particular, we test models with *sum* and *max* aggregators, as they are the most used in literature. As with PNA layers, we found that linear transformations are sufficient for $M$ and $U$ and, as in the original paper [17], we employ multiple towers.

$$X_i^{(t+1)} = U\left(X_i^{(t)}, \bigoplus_{(j,i) \in E} M\left(X_i^{(t)}, X_j^{(t)}\right)\right) \tag{14}$$

# G    Random graph generation

In this section, we present the details of the random generation of the graphs we used in the multi-task benchmark. Following previous work [18, 28], we opted for undirected unweighted graphs from a wide variety of types (we provide, in parentheses, the approximate proportion of such graphs in the benchmark). Letting $N$ be the total number of nodes per graph:

- **Erdős-Rényi** [39] (20%): with probability of presence for each edge equal to $p$, where $p$ is independently generated for each graph from $\mathcal{U}[0, 1]$

- **Barabási-Albert** [40] (20%): the number of edges for a new node is $k$, which is taken randomly from $\{1, 2, ..., N - 1\}$ for each graph

- **Grid** (5%): $m \times k$ 2d grid graph with $N = mk$ and $m$ and $k$ as close as possible

- **Caveman** [41] (5%): with $m$ cliques of size $k$, with $m$ and $k$ as close as possible

- **Tree** (15%): generated with a power-law degree distribution with exponent 3

- **Ladder graphs** (5%)

- **Line graphs** (5%)

- **Star graphs** (5%)

- **Caterpillar graphs** (10%): with a backbone of size $b$ (drawn from $\mathcal{U}[1, N)$), and $N - b$ pendent vertices uniformly connected to the backbone

- **Lobster graphs** (10%): with a backbone of size $b$ (drawn from $\mathcal{U}[1, N)$), $p$ (drawn from $\mathcal{U}[1, N-b]$) pendent vertices uniformly connected to the backbone, and additional $N-b-p$ pendent vertices uniformly connected to the previous pendent vertices.

Additional randomness was introduced to the generated graphs by randomly toggling arcs, without strongly impacting the average degree and main structure. If $e$ is the number of edges and $m$ the number of 'missing edges' ($2e + 2m = N(N - 1)$), the probabilities of toggling an existing and missing edge, respectively $P_e$ and $P_m$, are:

$$P_e = \begin{cases} 0.1 & e \le m \\ 0.1 \frac{m}{e} & e > m \end{cases} \qquad P_m = \begin{cases} 0.1 \frac{e}{m} & e \le m \\ 0.1 & e > m \end{cases} \tag{15}$$

After performing the random toggling, we discarded graphs containing singleton nodes, as they are in no way affected by the choice of aggregation.

## H  Graph type experiments

In order to better interpret the improvements in performance that the PNA brings, we tested the models against the various types of graphs in the multi-task benchmark. In particular, in these experiments, we trained the models on the whole dataset with the proportions described above and then tested them against datasets composed by just one category of graphs.

| Model | Erdos-Rényi | Barabási-Albert | Grid | Cave-man | Tree | Ladder | Line | Star | Cater-pillar | Lobster |
|---|---|---|---|---|---|---|---|---|---|---|
| PNA | -3.377 | -3.495 | -2.770 | -3.000 | -3.097 | -3.131 | -2.371 | -3.252 | -2.879 | -2.790 |
| MPNN-sum | -2.085 | -2.347 | -1.955 | -1.872 | -2.237 | -2.024 | -1.991 | -2.790 | -2.219 | -2.190 |
| MPNN-max | -2.807 | -2.943 | -2.383 | -2.523 | -2.484 | -2.721 | -1.980 | -3.066 | -2.379 | -2.339 |
| GAT | -2.361 | -2.578 | -2.111 | -2.027 | -2.161 | -2.250 | -1.892 | -2.678 | -2.134 | -2.114 |
| GIN | -1.840 | -2.084 | -1.769 | -1.679 | -1.912 | -1.842 | -1.672 | -1.927 | -1.913 | -1.877 |
| GCN | -1.930 | -2.187 | -1.740 | -1.536 | -2.039 | -1.841 | -1.691 | -2.088 | -1.997 | -1.974 |

Figure 8: Average $\log_{10}$MSE error across the various tasks of a particular model against a particular type of graphs.

The results, presented in Figure 8, show that the PNA improves across all types. However, it performs the worst on the graphs with a higher diameter (especially graphs close to lines), suggesting that the number of layers is not enough to reach the complete graph. Therefore, the main limitation to the PNA performance seems to be the message passing framework; this could motivate future research to try to improve the framework itself.

# I Standard architecture

In this section we will provide more intuition on the motivation behind our choice of architecture, presented in Section 3, which we will refer to as *recurrent*,[2] and present the results on a more *standard* architecture.

The main motivations behind the choice of the architecture were: (1) provide a fairer comparison between the models (2) showcase a parameter-efficient *recurrent* architecture with a prior [3] that works very well with the tasks at hand. In particular:

1. The GRU helps to avoid over-smoothing, and the models that do not have a skip connection across the aggregation (GAT, GIN and GCN) are those benefiting the most from it; therefore, to still provide a fair comparison in the results below, we added skip connections from every convolutional layer to the readout, in all the models.

2. The S2S (as opposed to a mean readout used below) helps the most architectures without scalers as it can provide an alternative counting mechanism.

3. The repeated convolutions are a parameter-saving prior which works well in these tasks but does not change the rank between the various models.

For completeness, we present in Figure 9 the comparison of the average results of the *recurrent* architecture and *standard* one which uses no GRU but skip connections, mean readout rather than S2S and a fixed number of convolutions (8).

| Framework | PNA | PNA no scalers | mean, max & min | MPNN sum | MPNN max | GAT | GIN | GCN |
|-----------|-----|----------------|-----------------|----------|----------|-----|-----|-----|
| Recurrent | -3.13 | -2.77 | -2.57 | -2.53 | -2.50 | -2.26 | -1.99 | -2.04 |
| Standard | -2.97 | -2.55 | -2.43 | -2.78 | -2.41 | -2.00 | -2.03 | -2.14 |

Figure 9: Average $\log_{10}$MSE error across the various tasks of a particular model when inserted in the *recurrent* or the *standard* model. The *mean, max & min* model represents a baseline MPNN which employs *mean*, *max* and *min* aggregators and no scaler.

# J Single task experiments

Apart from a good method to evaluate the performance on a variety of different problems, the multi-task approach offers a regularization opportunity that some models capture more than others. In particular, we found that models without scalers (or *sum* aggregator) are those benefiting the most from the approach; we hypothesise that the reason for this lies in some supervision that specific tasks give to recognise the size of a model neighbourhood. Moreover, more complex models are more prone to overfitting when trained on a single task. Figure 10 shows the average performance on the individual tasks of the various models.

| Framework | PNA | PNA no scalers | MPNN sum | MPNN max | GAT | GIN | GCN |
|-----------|-----|----------------|----------|----------|-----|-----|-----|
| multi-task | -3.13 | -2.77 | -2.53 | -2.50 | -2.26 | -1.99 | -2.04 |
| single task | -2.86 | -2.07 | -2.68 | -2.10 | -2.46 | -1.96 | -2.13 |

Figure 10: Average $\log_{10}$MSE error across the various tasks of a particular model either trained concurrently on all the tasks (*multi-task*) or trained separately on the individual tasks (*single task*). With the exception of the output layer, the two settings use the same architecture.

# K  Parameters comparison

Figure 11 shows the results of testing all the other models on the multi-task benchmark with increased latent size.

| Model | Size 16 | | Size 20 | |
|---|---|---|---|---|
| | # params | log score | # params | log score |
| PNA | 8350 | -3.13 | - | - |
| MPNN (sum) | 7294 | -2.53 | 11186 | -2.19 |
| MPNN (max) | 8032 | -2.50 | 12356 | -2.23 |
| GAT | 6694 | -2.26 | 10286 | -2.08 |
| GCN | 6662 | -2.04 | 10246 | -1.96 |
| GIN | 7272 | -1.99 | 11168 | -1.91 |

Figure 11: Average score of different models using feature sizes of 16 and 20, compared to the PNA with 16 on the multi-task benchmark. "# params" is the total number of parameters in each architecture.

We observe that, even with fewer parameters, PNA performs consistently better and an increased number of parameters does not boost the performance of the other models. This suggests that the multiple aggregators in the PNA produce a qualitative improvement to the capacity of the model.

## Footnotes

[2]Note that this was only used in the synthetic benchmarks, while in the real-world benchmarks, we kept the same architecture from Dwivedi *et al.*

[3]This prior corresponds to the knowledge that these tasks can be solved by the convergence of an aggregation function in the message passing context, potentially with an additional readout/function.