[Reviews · NeurIPS 2020]

Review 1

Summary and Contributions: This work is motivated by the (previously shown) fact that node message aggregation functions used in GNNs (sum, mean, min, max, standard deviation) are known to fail to distinguish between certain graphs. To address the limitation, this work proposes a framework, Principal Neighbourhood Aggregation (PNA), where multiple aggregators are combined along with a scaling term dependent on node degree. The use of use of multiple aggregators is theoretically justified as necessary for distinguishing between multisets (node neighborhoods) in continuous space. The specific aggregators considered in the PNA framework are mean, standard deviation, min, and max. The commonly-used sum aggregator is implicitly included since the scaling operators combined with the mean aggregation generalizes sum aggregation, as shown in the paper. Higher-order moment aggregators are mentioned as possibly useful in some graph domains, but are not included as part of the presented framework, empirically motivated (in the appendix) by the tendency for higher order moments to overfit, especially on small average degree graphs. Post-rebuttal update: I have read carefully the author rebuttal. They have provided new results, which they will include in the appendix, that address my concerns about the multi-task regularization effect and graph size. These additional results strengthen the submission.

Strengths: Novelty: The key contribution of the work is in the theoretical findings of the necessity of multiple aggregators for distinguishing between node neighborhoods for continuous input spaces. This finding dovetails from the Xu et al [1] finding, but the consideration of continuous input spaces is particularly valuable because intermediate latent node features in deep GNN models exist in such continuous spaces. Understanding the representation power of GNN models for these latent features is crucial in gaining an understanding of the models. The idea of combining multiple aggregators is not novel. The paper correctly identifies an example of combined aggregation functions in Dehmamy et al [2]. However, the particular theoretical development as it relates to the necessity for multiple aggregator functions (Theorem 1) and aggregators based on the moments are new in this work, as well as the use of mean aggregation combined scaling based on neighborhood size. Significance and Impact: The idea of aggregating and scaling node messages is general enough that if can be applied to most GNN architectures, and some (GCN, GAT, GIN, and MPNN) are indeed considered in the work. The theoretical findings have implications across all of these architectures that consist of aggregating neighborhood messages throughout a graph. [1] Xu, Hu, Leskovec, Jegelka. How Powerful are Graph Neural Networks? arXiv preprint arXiv:1810.00826, 2018 [2] Demamy, Barabasi, Yu. Understanding the representation power of graph neural networks in learning graph topology. NeurIPS 2019.

Weaknesses: Methodological: The work here places importance on topology/structure. For example, the message scaling is dependent on node degree. Thus this method is apt for applications where the structure is paramount, e.g. one such application mentioned is reasoning about social networks where the degree of the nodes/users provides a lot of information about that node/user. Though useful in many domains, there are domains where GNNs are useful but topology is not important. This is reflected empirically for regular grid graph of the computer vision datasets where PNA does not significantly improve over other methods. Empirical: * The training on multiple tasks may have a regularizing effect that is not discussed in the work. Results on individual tasks may be useful to gain some insights into the power of the method on these data. * The synthetic dataset is comprised of a nice variety of graph types, but results are reported for all of them. It is not clear how much the framework improves on tasks on different graph types. Splitting these results, perharps in the appendix may be useful for a more nuanced insight into the benefits of PNA * Only undirected and unweighted graphs are considered, excluding many important graph applications eg computation graphs, nonsymmetric social networks * The empirical analysis on synthetic data only considered relatively small graphs (15-50 nodes) which do not capture the scale of many real-world graph applications. The real datasets considered (molecules, CIFAR10, and MNIST) are limited. For example the computer vision datasets are regular grids, where the topology is less important. Indeed, the results show on those datasets (no significant improvement over SotA) highlight the fact that the representation shortcomings addressed in this work are in distinguishing *structure* in the underlying graphs, the importance of which are application dependent. * Though the synthetic dataset is highlighted as one of the contributions, the novelty and impact of this particular contribution is quite secondary. Random graphs are commonly used to evaluate GNN models and the particular tasks on these graphs considered here are tasks of identifying different graph properties. Though sound and necessary for evaluation, it is not a particularly noteworthy contribution. To be clear, the paper contribution does not hinge on this anyway.

Correctness: The theorems and propositions appear to be correct following the proofs provided in the appendix. The empirical methodology mostly follows conventions in the field for evaluating GNNs

Clarity: The paper is well written and the motivation and development are clearly structured

Relation to Prior Work: Prior work on empirical and theoretical developments in graph representation power are correctly cited and the connection to the most closely related works are clearly discussed

Reproducibility: Yes

Additional Feedback:


Review 2

Summary and Contributions: The paper proposed Principal Neighbourhood Aggregation (PNA) as a new aggregator for Graph Neural Networks. Principal Neighbourhood Aggregation combines multiple existing GNN aggregators and achieve superior performance. ------------------------ Updates: The authors provide additional experimental results in the response, which answers my questions. Overall, I like the technical contribution, although the experiments can be improved in the original manuscript. With these new results, I'm happy to vote for acceptance.

Strengths: The proposed approach is technically sound. The experimental results are strong under the proposed experimental settings.

Weaknesses: The paper does not report the performance on standard GNN benchmarks, such as node classification on Cora and graph classification on TU dataset. These evaluations are simple to conduct, and the authors should at least give a brief discussion. The GNN model used is also not standard. The paper uses a GNN with recurrent GNN layers, where weights are shared across different layers. This kind of GNN has less representation power, and it is likely that under this setting, the proposed Principal Neighbourhood Aggregation has clearer advantage as it uses an ensemble of aggregators. I think using standard GNNs where different layers do not share weights would make the experimental results more convincing. Important baselines are missing. It would be really helpful to show the performance of GNNs with combinations of max, min, mean aggregators that are commonly used in the current literature, without the proposed Normalized moments aggregation and Degree-based scalers.

Correctness: Yes.

Clarity: Yes.

Relation to Prior Work: Yes.

Reproducibility: Yes

Additional Feedback: Overall, the paper works on an interesting direction on improving GNN models, that is to use more sophisticated aggregation functions. However, I think the current evaluation is not convincing and can be greatly improved. Besides the comments I mentioned above, here are my additional comments. (1) How does Principal Neighbourhood Aggregation compares with a naive baseline, where max, mean and min aggregators are used? This baseline should presumably work better than only using one aggregator, however the paper does not make a comparison. This important baseline will greatly help justifying the proposed method. (2) The GNN model as well as the evaluation setup are not standard. The proposed aggregation function should be orthogonal to the GNN architecture and the dataset, so it is not clear why standard setups are not used.


Review 3

Summary and Contributions: This work improves the original GIN model in multiple ways; generalizing the injective sum aggregation function to more expressive multiple aggregators, a new architecture called Principal Neighbourhood Aggregation combining graph convolution and recurrent mechanism, and learning on multiple tasks simultaneously.

Strengths: It is an important modelling contribution toward more expressive aggregation operations for GNNs. Numerical results are also convincing.

Weaknesses: Question on baseline: What is the performance of the simplest architecture using only the new multiple aggregators defined in Eq.8? That is, a CNN-like architecture that stacks M x GC layers (each layer has its own weights) with no GRU and no S2S in Fig.2? An alternative question is : Are GRU and S2S critical for good performances? And if yes, why?

Correctness: Sounds correct.

Clarity: Yes, the paper is nicely written.

Relation to Prior Work: Yes, great work.

Reproducibility: Yes

Additional Feedback:


Review 4

Summary and Contributions: The authors propose a new aggregation method for GNNs. They theoretically prove that mixing multiple aggregation methods performs better than a single one. They also introduce moment aggregators. Empirical results show that the proposed method outperforms existing methods.

Strengths: The paper addresses an important problem in GNNs, aggregation of information from neighboring nodes. They theoretically prove that using multiple aggregators the performance of GNN could be improved if the node features are continuous. The proofs seems to be correct and the experiments are carried out well showing improvement compared to baselines. There are also insightful ablation studies in the paper and supplement.

Weaknesses: The proposed method is indeed novel in geometric deep learning. However, the idea of mixed pooling [1] has already been explored in CNNs. Although there are obvious differences, they are in the same spirit as the proposed method. It would be interesting to add a few ablation studies: 1- A more comprehensive ablation study (besides the one in the supplement) on the number of moments needed (preferably on real-world datasets such as social network as mentioned in the paper). Is it linearly dependent to the average degree? As mentioned in the supplement, max and min seems to be more consistent aggregators. Could they reduce the number of required moments (for example to sub-linear)? 2- Which of these 4 aggregators in PNA is more important? Are they equally important? [1] Generalizing Pooling Functions in Convolutional Neural Networks: Mixed, Gated, and Tree, Lee et. al., AISTAT 2016.

Correctness: The claims seems to be theoretically correct.

Clarity: The paper is well written and clearly organized.

Relation to Prior Work: Relation to prior works have been adequately discussed.

Reproducibility: Yes

Additional Feedback: ==== post rebuttal ==== I have read the authors' response as well as other reviews during the discussion period in August. I appreciate their efforts and I keep my score as is.

[Author Response · NeurIPS 2020]

Firstly we would like to thank all the reviewers for their very insightful comments and suggestions. We hope these points appropriately address the reviewers' concerns, and they will be incorporated in more detail in the paper.

**Baseline with mean, max and min aggregators.** As Reviewer 2 suggested, we have added the PNA without *std* and scalers to the results below to better highlight the improvement brought by those components which are, to the best of our knowledge, entirely novel in the graph machine learning literature. As expected, its performance lies between the model with also the *std* (PNA no scaler) and the one with just *max* aggregator (MPNN max).

**Most important aggregator.** Answering to Reviewer 4, experiments showed that the choice of aggregator is very much task-dependent, e.g., for the graph theory artificial dataset (Figure 3) we found the *mean* was the best performing aggregator, whereas in computer vision tasks (Figure 5) is the *max* aggregator. The result achieved in tasks where we found out one aggregator was significantly more important than the others may suggest the PNA is able to focus on such aggregator.

**Structure of the GNNs.** We understand Reviewers 2 and 3's concern with the non-standard architecture using GRU, S2S, and repeated convolutions. We will clarify in the paper that (1) this is only used in the synthetic benchmarks, while in the real-world benchmarks, we kept the same architecture from Dwivedi *et al.*, (2) this architecture was chosen to provide a fairer comparison between the models as later explained. However, for completeness, we reckoned it important to run the models on a standard GNN architecture and report the results below. The GRU helps to avoid over-smoothing, and the models that do not have a skip connection across the aggregation (GAT, GIN and GCN) are those benefiting the most from it; therefore, to still provide a fair comparison in the results below, we added skip connections from every convolutional layer to the readout, in all the models. The S2S (as opposed to a mean readout used in the results below) most helps architectures without scalers as it can provide an alternative counting mechanism. Finally, the repeated convolutions are a parameter-saving prior which works well in these tasks but does not change the rank between the various models. We will clarify better the thought process behind choosing the architecture and add these results in the appendix to address these types of concerns in our final version.

**Single task results.** As Reviewer 1 correctly suggested, the multi-task approach offers a regularization opportunity that some models capture more than others. In particular, we found that models without scalers (or *sum* aggregator) are those benefiting the most from the approach; we hypothesise that the reason for this lies in some supervision that specific tasks give to recognise the size of a model neighbourhood. Moreover, more complex models are more prone to overfit when training on a single task. Due to space limitations, we only report the average performance, the detailed per-task performance and analysis will be added to the appendix of the paper.

| Framework | PNA | PNA no scalers | mean, max & min | MPNN sum | MPNN max | GAT | GIN | GCN |
|---|---|---|---|---|---|---|---|---|
| multi-task | -3.130 | -2.770 | -2.570 | -2.530 | -2.500 | -2.260 | -1.990 | -2.040 |
| multi-task standard | -2.970 | -2.550 | -2.430 | -2.780 | -2.410 | -2.000 | -2.030 | -2.140 |
| single task | -2.860 | -2.070 | -1.850 | -2.680 | -2.100 | -2.460 | -1.960 | -2.130 |

**Graph type results.** Following the suggestion by Reviewer 1, we have tested the models' performance across the various types of graphs in the synthetic benchmark. The results show that the PNA improves across all types; however, it performs the worst on the graphs with higher diameter (especially graphs close to lines), suggesting that the number of layers is not enough to reach the complete graph. Therefore, the main limitation to the PNA performance seems to be the message passing framework; this could motivate future research to try to improve the framework itself.

| Model | Erdos-Rényi | Barabási-Albert | Grid | Cave-man | Tree | Ladder | Line | Star | Cater-pillar | Lobster |
|---|---|---|---|---|---|---|---|---|---|---|
| PNA | -3.377 | -3.495 | -2.770 | -3.000 | -3.097 | -3.131 | -2.371 | -3.252 | -2.879 | -2.790 |
| MPNN-sum | -2.085 | -2.347 | -1.955 | -1.872 | -2.237 | -2.024 | -1.991 | -2.790 | -2.219 | -2.190 |
| MPNN-max | -2.807 | -2.943 | -2.383 | -2.523 | -2.484 | -2.721 | -1.980 | -3.066 | -2.379 | -2.339 |
| GAT | -2.361 | -2.578 | -2.111 | -2.027 | -2.161 | -2.250 | -1.892 | -2.678 | -2.134 | -2.114 |
| GIN | -1.840 | -2.084 | -1.769 | -1.679 | -1.912 | -1.842 | -1.672 | -1.927 | -1.913 | -1.877 |
| GCN | -1.930 | -2.187 | -1.740 | -1.536 | -2.039 | -1.841 | -1.691 | -2.088 | -1.997 | -1.974 |

As Reviewer 1 highlighted, indeed, the multi-task benchmark is undoubtedly not the main contribution of our paper; we will try to clarify that is not the case, but instead to motivate our need for the creation of this flexible benchmarking tool.

**Standard datasets.** We had initially omitted considering Cora and other datasets given their known oversaturation and the lack of potential for comparing existing GNN approaches. As recommended by Reviewer 2, we have conducted preliminary tests on them and found that the results are consistent with the existing state-of-the-art ($> 85\%$ on Cora). We will incorporate this discussion within our paper.

Finally, we want to thank Reviewer 4 for bringing to our attention the interesting work by Lee *et al.* on mixed pooling in computer vision. Although in a different field, the motivations behind it are similar to those that led us to this work; therefore, we will add a discussion of the connection between the two fields in our introduction.

[Meta-Review · NeurIPS 2020]

All four referees expressed enthusiasms on this paper during review and discussion. Therefore, an accept is recommended.